# A Comparative Study of PMETAC-Modified Mesoporous Silica and Titania Thin Films for Molecular Transport Manipulation

**DOI:** 10.3390/polym14224823

**Published:** 2022-11-09

**Authors:** Sebastian Alberti, Juan Giussi, Omar Azzaroni, Galo J. A. A. Soler-Illia

**Affiliations:** 1Department of Physics and Technology, UiT the Arctic University of Norway, NO-9037 Tromsø, Norway; 2YPF-Tecnología S.A, Berisso 1925, Argentina; 3Instituto de Investigaciones Fisicoquímicas Teóricas y Aplicadas (INIFTA), Departamento de Química, Facultad de Ciencias Exactas, Universidad Nacional de La Plata-CONICET, La Plata 1900, Argentina; 4Instituto de Nanosistemas, Universidad Nacional de General San Martín-CONICET, San Martín 1650, Argentina

**Keywords:** hybrid materials, mesoporous thin films, molecular brushes, permselective membranes, zwitterionic

## Abstract

The manipulation and understanding of molecular transport across functionalized nanopores will take us closer to mimicking biological membranes and thus to design high-performance permselective separation systems. In this work, Surface-initiated atom transfer radical polymerization (SI-ATRP) of (2-methacryloyloxy)-ethyltrimethylammonium chloride (METAC) was performed on both mesoporous silica and mesoporous titania thin films. Pores were proven to be filled using ellipsometry and diffuse reflectance infrared Fourier transform spectroscopy (DRIFTS). Furthermore, the employed method leads to a polymer overlayer, whose thickness could be discriminated using a double-layer ellipsometry model. Cyclic voltammetry experiments reveal that the transport of electrochemically active probes is affected by the PMETAC presence, both due to the polymer overlayer and the confined charge of the pore-tethered PMETAC. A more detailed study demonstrates that ion permeability depends on the combined role of the inorganic scaffolds’ (titania and silica) surface chemistry and the steric and charge exclusion properties of the polyelectrolyte. Interestingly, highly charged negative walls with positively charged polymers may resemble zwitterionic polymer behavior in confined environments.

## 1. Introduction

Molecular transport through nanoporous materials is a fascinating subject that has gained relevance during the last decade. Mesoporous materials with highly controlled area-to-volume ratio and confinement effects are an excellent choice for fundamental and applied studies in this subject. They have proven to be ideal as adsorbents or for sensing applications [1], and in the last decade, a plethora of new mesoporous hybrid nanosystems have been designed and produced, with a variety of applications in fields as diverse as controlled release, heterogeneous catalysis, hydrogen energy or solar cells [2,3]. Moreover, mesoporous materials have been considered as a model system for fundamental physicochemical studies on transport and confinement processes, which are of paramount importance in membranes for electrophoresis and nanofluidics [4,5]. Achieving complete understanding of the synthesis and transport properties will take us one step closer to mimicking biological membranes and enzymatic cavities, leading to predesigned bioinspired nanosystems with programmable features [3,6].

Sol–gel chemistry provides a versatile tool for inorganic oxide thin films and particle synthesis capable of finely tuning the pore size and pore connectivity [7], crucial features in their applications such as drug delivery, permselective membranes and optical coatings. Additionally, the isoelectric point of the membrane or particles can be carefully selected by the correct choice of precursors. In this manner, we can study and optimize the main transport restrictions, charge and steric hindrance. Nevertheless, the possibility of tunning the membrane charge density or polarity as well as wetting properties without modifying the porous structure is essential to achieve a successful design. In this regard, polymer grafting to mesoporous inorganic oxide thin films used as scaffolds is an excellent approach to reach and improve those objectives.

Polymer chemistry has expanded our ability to conceive and synthesize hybrid materials. The large availability of monomers and techniques has provided unlimited possibilities for further tailoring and diversifying hybrid materials. In particular, polymer brushes have prompted an interest in the design of polymer thin films with strict molecular control and new flanged structures that exhibit better performance for specific applications [8,9]. In this regard, a wide palette of polymer brushes has been designed and synthetized, and subsequently used to attain a large variety of physical and chemical conformational changes in response to diverse stimuli [3,10,11,12]. Coupling polymer brushes to porous materials proved to be a robust way of changing transport behavior, as this approach not only allows us to change the confined species but may also be able to introduce double responsiveness and higher complexity to the system regardless of the external medium. Particularly, polyelectrolyte brushes, due to their highly charged nature, are appropriate molecules to introduce or change the surface charge, and therefore are able to tune the transport behavior of charged probes for rectification, permselectivity or ion responsiveness purposes [13,14,15,16]. Therefore, a positive polyelectrolyte with quaternary ammonium groups, PMETAC, was chosen to functionalize titania and silica mesoporous scaffolds.

In this work, two mesoporous inorganic oxides with different isoelectric points were modified by post-grafting techniques and polymerization of a positively charged monomer (METAC) through an ATRP process [17,18,19]. This process leads to a complete mesopore filling and a thin polymeric brush overlayer on top of the film surface, as proven by ellipsometric measurements. The nature of the inorganic layer affects polymer brush growth both inside and outside the porous structure. Finally, the influence of PMETAC on transport properties for negative and positive probes permitted the identification of two exclusion mechanisms that take place simultaneously, due to steric hindrance and charge exclusion effects. The synergy between the polymer charge and the surface charge of the inorganic titania or silica matrices was considered in the analysis. This work provides new experimental data that could be compared to computational simulations advances to allow a better insight into the permselective predesigned properties of mesoporous hybrid materials.

## 2. Materials and Methods

### 2.1. Synthesis

The synthesis of mesoporous thin films was conducted as described in previous works [20,21]. The oxide precursors (tetraethoxysilane (TEOS)—Merck (Buenoss Aires, Argentina) and TiCl_4_—Sigma-Aldrich (Buenoss Aires, Argentina)) were used in alcohol–water solutions, in the presence of a supramolecular template (F127 block copolymer, Aldrich, M = 13,600). Silica films were prepared from a sol of molar ratios TEOS:EtOH:H_2_O:HCl:F127 = 1:40:10:0.008:0.005, which was aged at room temperature for 3 days, in order to optimize the hydrolysis and condensation processes. Titania films were prepared from fresh solutions of molar ratio TiCl_4_:EtOH:H_2_O:F127 = 1:40:10:0.005. These precursor solutions were used to produce films by dip coating onto glass (Marienfeld, Lauda-Königshofen, Germany) or ITO substrates (ITO, Delta Technologies, Buenos Aires, Argentina, R = 8–12 Ω sq). Silica films were prepared under 50% relative humidity conditions at 25 °C, and 3 mm s^–1^ withdrawal speed, and titania thin films under 30% relative humidity and at the same withdrawal speed. The organic template was removed by calcinations at 350 °C after heat treatments at 60 °C for 30 min and other 30 min at 130 °C. Post-grafting was performed using the amine precursor 3-aminopropyltriethoxysilane (APTES, Sigma-Aldrich, 97%) though chemical vapor surface modification at ambient temperature for 10 min under nitrogen atmosphere. Briefly, a film sample was sealed in a proper flask with a septum and purged with dry nitrogen for 15 min before a nitrogen atmosphere saturated in APTES was flowed through. Subsequently, the surface modification of the mesoporous amino–silica film with the initiator groups was accomplished by the following procedure: 0.4 g (1.7 mmol) of 2-bromoisobutyryl bromide (BIB) and 0.92 g (4.5 mmol) of triethylamine (Et_3_N) in methylene chloride (Anedra, Buenos Aires, Argentina, 100% purity) were added to a single-neck Schlenk flask and closed with a rubber septum. The reactants were degassed under vacuum for 30 min followed by backfilling with N_2_(g). Mesoporous films were sealed in Schlenk tubes and degassed (4× high-vacuum pump/N_2_ refill cycles). The reaction mixture was syringed into the Schlenk flasks containing the mesoporous films and left overnight under N_2_(g) at room temperature. The modified substrates were then washed twice with methylene chloride followed by washing with ethanol.

Once the initiator was anchored, the functionalized mesoporous film was directly used as a substrate for ATRP. Poly(2-methacryloyloxy)ethyltrimethylammonium chloride (PMETAC) brushes were grown on initiator-functionalized mesoporous silica films. In a 100 mL Schlenk flask, 23.3 g of (2-methacryloyloxy)ethyltrimethylammonium chloride as 80 wt% solution in water was added. The monomer solution was diluted with methanol to 50 wt%. CuBr and Bipyridine (bipy) were added. This solution was degassed by N_2_(g) bubbling for 1h. The initiator-functionalized single mesoporous silica substrates were sealed in a Schlenk flask and degassed (4× high-vacuum pump/N_2_ refill cycles). The degassed monomer solution was syringed into this Schlenk flask until all substrates were entirely covered. Mesoporous substrates were removed from the polymerization solution after 20 h, thoroughly washed with water, and dried to yield poly(2-methacryloyloxy) ethyltrimethylammonium chloride brush-grafted mesoporous films [9,22,23]. This procedure was conducted in the same way for both silica and titania MTF.

### 2.2. Thin-Film Characterization

Diffuse reflectance infrared Fourier transform spectroscopy. Measurements (DRIFTS) were performed on a Nicolet Magna 560 instrument, equipped with a liquid nitrogen-cooled MCT detector. DRIFTS measurements were performed by depositing scratched film samples on a KBr-filled DRIFTS sample holder.

Environmental ellipsometric porosimetry (EEP). Film thickness and refractive index in the 200–950 nm region were obtained using a SOPRA GES5A spectroscopic ellipsometer using microspot configuration. Measurements of the ellipsometric parameters Ө (Theta) and Δ (Delta) were carried out under dry nitrogen flux in order to avoid water condensation within the mesopores. The film refractive index was satisfactorily adjusted according to a single-layer model for the amino-functionalized thin films. A double-layer model was adjusted for the polymer-modified mesoporous films and compared to the single-layer model in order to verify the need to increase modeling complexity. Refractive index and film thickness for mesoporous and polymer layers in a double-layer polymer were adequately adjusted. A Brüggemann effective medium approximation (BEMA) was used to calculate the pore volume fraction (V_pore_), considering two phases made up of silica (n_633_ = 1.455)/titania (n_633_ = 2.15) and void pores [24]. The polymer volume fraction in the material V_PMETAC_/% was obtained by analyzing the optical data with a three-component BEMA using n_633_ = 1.455 or n_633_ = 2.15 for the silica and titania framework, respectively, n_633_ = 1.56 for PMETAC, and n_633_ = 1 for void pores. Water adsorption–desorption curves (at 298 K) were measured by environmental ellipsometric porosimetry (EEP, SOPRA GES5A). Film thickness and refractive index values were obtained from measuring the ellipsometric parameters, according to the method reported by Boissière [25]. Pore size distributions were obtained from the analysis of the refractive index variation along the variation of the water vapor pressure, using WinElli 2 software (SOPRA, Inc., Paris, France).

Cyclic voltammetry. Measurements were performed using a potentiostat TQ 03 with an Ag/AgCl reference electrode. All probe solutions were prepared with a concentration of 1 mM in 100 mM KCl as supporting electrolyte resulting in a pH 5–6 solution. Qualitative variations in permselectivity were studied by following the changes of the voltametric peak currents associated to cationic [Ru(NH_3_)_6_]^2+/3+^ and anionic [Fe(CN)_6_]^4−/3−^ redox probes that diffused across the mesoporous film [26,27,28,29,30].

Scanning Electron Microscopy (SEM). Images were recorded to assess the mesoporous thin film deposition and the porous structure and polymer deposition. A Zeiss SUPRA 40 field-emission scanning electron microscope instrument was used to this end.

Small-Angle X-Ray Scattering (SAXS). Patterns were acquired at the Austrian SAXS beamline at ELETTRA synchrotron (Trieste, Italy), using a 1.54 Å (8 keV) incidence X-ray beam [31]. The sample was placed at 82.88 cm from a pixel detector (Pilatus 1M) on a rotation stage, which allowed a glancing angle to be set between the incident radiation and the sample to 3°. The samples were prepared on coverslips (ca. 0.15 mm thick) to allow measurements in Laue geometry. The angular scale of the detector was calibrated using Ag-behenate as a reference pattern. 

## 3. Results and Discussion

Mesoporous silica and titania thin films (MTF-TiO_2_/MTF-SiO_2_) were produced by dip coating on glass and indium tin oxide substrates using Pluronics F127 block copolymer as the structure-directing agent. After deposition, transparent mesostructured hybrid phases composed of inorganic building blocks and an entrapped organized supramolecular template presented periodicity at a 10 nm scale as a consequence of the supramolecular templating effect. This step was followed by template removal, which led to the actual mesoporous material (Figure 1) [1,32]. The formation of this first mesostructured hybrid phase was critical to obtain a final mesoporous material with tailored features in a reproducible way. Mesoporous thin films have tunable, stable and rigid structures. 

Calcination of the samples allowed for a more stable and rigid structure and guaranteed complete template removal, as demonstrated by SEM and SAXS characterization (Figure 2a,b). Subsequently, amino groups were grafted to the silica and the titania matrix by a post-grafting technique. The post-grafting process was compatible with previously calcined substrates and created a higher density of available chemical groups as compared to co-condensation [33]. This could also be confirmed by measuring the monolayer of APTES and 2-bromoisobutyryl bromide in a planar surface of silicon, being (0.6 ± 0.1) for APTES and (0.4 ± 0.1 nm) for BIB. A comparison of monolayer thickness and the decrease in pore volume confirms 2.3–3 molecules per nm^2^ for silica and 1.5–2 for titania. Consequently, a high number of grafting sites for the ATRP initiator (at least 90% of these grafting sites reacted with 2-bromoisobutyryl bromide) and subsequent polymerization were available when compared to co-condensation synthesis of thin films [34,35]. Post-grafted films were carefully analyzed by EEP due to the possibility of pore occlusion or an inhomogeneous distribution of the desired molecules. To rule out this possibility, control measurements were taken at each functionalization step. Water adsorption–desorption isotherms measurements using EEP are shown in Figure 2c,d. Pore volume can be directly extracted from the refractive index changes in the isotherms, as air (refractive index 1) is replaced by water (refractive index 1.33) inside the pores. In APTES-mesoporous films, EEP shows that hysteresis takes place, which is a feature of mesoporous films, and a pore volume of 31% and 28% (silica and titania, respectively) was attained and a thickness of the layer was 150 and 180 nm, respectively. This control measurements show a decrease in pore volume, corresponding to a high-density post-grafting functionalization without pore blocking, and disregards any unspecific adsorption of monomers. This is a key feature to achieve complete pore filling by polymer brushes. Even though silica has been the most used scaffold for functionalization in the design of drug carriers and permselective membrane, here we prove that titania can be functionalized in an analogous way. The possibility to functionalize titania brings advantages due to its performance against mechanical weathering and dissolution in aqueous environments [2,36,37].

Once the initiator was anchored, the functionalized mesoporous film was directly used as a substrate for the ATRP process in methanol as a solvent. Both titania as well as silica films were functionalized for comparison. The presence of polymer brushes on both systems was proven by comparing DRIFTS and EEP of the samples after ATRP, the initial mesoporous material and the APTES-functionalized controls (Figure 3 and Figure 4). Figure 3 depicts DRIFT spectra of amino-functionalized films (black full line) compared to post-polymerization samples (blue full line) for both titania and silica thin films. Inorganic framework bands corresponding to Si-OH and Si-O-Si stretching at 966 and 1080 cm^−1^ respectively, Ti-O-Ti modes at 860 and Ti-OH at 750 cm^−1^ did not change significantly upon polymerization. A shift was observed in the characteristic bands corresponding to the N-H asymmetric bending of the amine group around 1560 cm^−1^, indicating changes in the amine groups. In addition, upon polymer modification, a C=O stretching bond at 1727 cm^−1^ appears, which is attributable to the ester linkage of METAC monomer units in the PMETAC polymer backbone. The DRIFTS measurements indicate that PMETAC is formed in the mesoporous films under the explored conditions, as could be expected from previous work [38]. However, this technique alone cannot provide an assessment of whether the polymer is located within the mesopores or on the film surface.

To evaluate this central aspect, EEP was performed on PMETAC-containing film samples. To satisfactorily adjust the ellipsometric parameters between 300 and 900 nm, a two-layer model was proposed to analyze the changes in refractive index and thickness of PMETAC-containing films (Figure 4). The layer close to the substrate represents the mesoporous thin film, and the topmost layer represents the polymer brush grown on the surface. This double-layer model was compared to a single-layer model commonly used in previous reports [26,39] and repeated in this work for comparison (Appendix A). Three consistent and marked differences are observed with respect to the EEP curves for the films shown in Figure 1: (a) the refractive index at P/P° = 0 increases, (b) accessible pore volume decreases significantly, (c) no significant adsorption–desorption hysteresis is observed (Figure 4). These findings are consistent with a practically complete pore filling with polymer, as the higher refractive index at 0% humidity is due to the replacement of air with polymer brushes and initiator molecules. Decreases in volume fraction of 15% and 13% for silica and titania, respectively, after polymerization allow us to estimate the number of monomers per surface area, being 2 nm^−2^ for titania and 3 nm^−2^ for silica, from molecular volume values taken from the literature and geometrical considerations [24,40,41,42]. If an initiation efficiency is assumed to be 35 % as reported elsewhere, a degree of polymerization (DP) of 2–3 would be expected for both titania and silica [22]. Even though the pores were completely filled, this quantity of monomers was less than the maximum expected for a 7 nm diameter pore due to loss of volume after the post-grafting procedure. In addition, after ATRP, mesoporosity turns to microporosity, a feature in line with a significantly lower pore volume (approximately 7%) as compared to the unmodified mesoporous film. The pore and neck size distributions were calculated according to the model proposed by Boissière et al. for ellipsoidal mesopores [25]. Pore diameter changed from 7 nm to 2 nm and necks from 4 nm to 1 nm after ATRP functionalization. The absence of hysteresis after polymerization can be compared to as synthesized MTF prior calcination, where pores are still filled with the template (Appendix A). It is worth mentioning that the pore size distribution model uses contact angle measurements to gauge the interactions of water and oxide surfaces, which strongly influence the obtained pore diameter and can lead to misleading conclusions, especially if small differences are analyzed. Still, the thickness and shape of the sorption isotherms are independent of the assumptions done by Boissière model [25]. The results here reported are in line with previous reports for mesoporous silica, and confirm that the double-layer model is an indispensable tool for studying polymer content in nanometric cavities [39,42].

The top layer of this model represents the polymer brush on the surface. Its thickness value at P/P° = 0 was calculated to be 12 nm for titania and 20 nm for silica (it needs to be noted that the APTES and the initiator layer thickness are not modeled), corresponding to a DP of 30 and 70, in agreement with previous reports [22,43]. As relative humidity increases, the thickness of the polymer overlayer does too, while its refractive index decreases due to swelling with condensed water, as reported previously for polymer layers and hydrogels [44]. Consequently, the surface polymer layer practically doubles its thickness due to water uptake at high relative humidity values and decreases its refractive index below (1.45 ± 0.02) from a value of (1.56 ± 0.02). Despite the error of the measurement due to correlation between parameters, the value remains slightly higher than previously reported for PMETAC (1.51 corrected for dry layer) and can be subject to variations due to the density of the polymer brush layer (ATRP results in higher polymer density) and counter ions available (Cl^−^, Br^−^) [39] or non-terminated chains. Correlation between parameters for layers below 20 nm has already been a challenging subject even on standard substrates, and an additional porous layer increases the complexity of the model [45,46]. To minimize the number of parameters, the substrate (ITO on glass) was previously modeled and its parameters kept constant. The mesoporous layer thickness was fixed due to minimal expected changes with humidity in microporous hybrid material [25]. To give further support to the results, additional measurements were performed for comparison. Appendix A shows the EEP curves obtained using a single-layer model for the hybrid film. After a smooth refractive index increase at low P/P° values, a decrease is obtained for higher relative humidity, altogether with an increase in the film thickness. This behavior has no physical meaning, which demonstrates that care must be taken in proposing consistent models for EEP experiments when dealing with complex systems. Furthermore, as synthesized MTF prior to calcination was studied by ellipsometry, in this scenario, pores are filled with the copolymer template but no polymer brush is present on the top surface (Appendix A), and no significant swelling is evident at high humidity. The two-layer model proposed here thereby provides a complete explanation of the changes observed when the films are exposed to increasing relative humidity: the refractive index increase on the mesoporous layer along water adsorption accompanied by both a decrease in the refractive index and an increase in the thickness of the polymer overlayer. The model is more suitable and gives a physical explanation for these systems, despite a greater complexity. The analysis of refractive index changes to distinguish pore blocking from pore filling is usually omitted in the literature and prior reports have not provided a clear distinction between polymers on top and inside pores. We proved that polymer brushes on top of the thin films present a higher PD than the polymer chains inside the pores, which are merely oligomers. Brunsen and Silies showed that PD is significantly lower in polymers inside confined spaces as compared to polymers on the surface [41,42]. Therefore, employing a single-layer model in this case would overestimate the quantity of monomers inside the pores.

Contact-angle measurements (see Appendix A) confirm the film functionalization results obtained by the techniques discussed above. The unmodified mesoporous silica, a very hydrophilic film, exhibited a small contact angle of 5°. Upon APTES and APTES-polymer initiator modification, the contact angle increased significantly up to 45°. This increment in hydrophobicity is evident in the displacement of the hysteresis to higher humidity (Figure 2c,d). On the contrary, after PMETAC brushes were grown from the films’ surface, the contact angle decreased to 13°. This value matches the literature estimates [47]. Although titania samples show a slightly more hydrophobic behavior than silica, displaying higher contact angles, a similar trend was registered in the changes upon the addition of APTES, initiator and PMETAC polymerization.

Electrochemical measurements were performed in order to assess the transport of electrochemical probes across the mesoporous films. Cyclic voltammetry (CV) is a widely used technique to study pore accessibility. A faradic current will only be registered when electrochemical probes reach the electrode below the mesoporous coating. The signal may be enhanced by adsorption and preconcentration, or diminished by steric hindrance or charge exclusion, as shown in previous works [37]. Despite the fact that quantitative results are difficult to obtain due to a variety of electron transfer pathways that take place, this is a suitable and straightforward technique to obtain qualitative and semiquantitative information. To study transport properties through porous silica and titania, ferricyanide, [(Fe(CN)_6_]^3−/4−^ and hexaammineruthenium, [Ru(NH_3_)_6_]^3+/2+^ were chosen as negatively and positively charged probe molecules, respectively. These probes were either prompted to be excluded or adsorbed by the negatively charged matrix walls. Previous works using PMETAC-modified mesoporous silica thin films with variable polymer content had shown that a gradual variation of ionic permselectivity can be tuned for increasing PMETAC loadings within the mesopores. The interplay between silanol sites and PMETAC groups leads to a complex transport behavior, which has to take into account electrostatic factors such as exclusion or preconcentration [41], as well as specific adsorption of electroactive species on the pore surface that might lead to a percolative hopping pathway [37,48]. Figure 5 shows typical CV results of the negatively or positively electrochemical probes [(Fe(CN)_6_]^3-^) and [Ru(NH_3_)_6_]^3+^, respectively [49]. As shown in Figure 5, silica mesoporous film walls bear a slightly negative charge at pH 3, while they are heavily charged at pH > 4.5, due to the deprotonation of the surface silanol groups, leading to exclusion and preconcentration (adsorption) of charged analytes depending on their individual charges. It can be clearly seen that [Ru(NH_3_)_6_]^3+/2+^ is not only allowed to reach the electrode, but is preconcentrated by adsorption in both oxide films, which explains the large current densities in the order of 200–400 µA cm^−2^ and the distorted figure due to a convoluted signal of diffusive and adsorptive behavior (Figure 5a). The presence of this adsorption phenomenon is additionally supported by its dependence on ionic strength and the increasing signal registered during the first 5 min of cycling as previously reported for silica and silica-zirconia-based mesoporous thin films reported by Alberti et al. [37] On the other hand, the [(Fe(CN)_6_]^3−/4−^ probe is practically excluded in silica, but a signal is recorded in the titania films (Figure 5b). This difference can be attributed to the relevant role of the electrostatic attraction or repulsions, combined with the different isoelectric points of silica and titania, which are around pH 2 and 5, respectively [50]. Silica surfaces are negatively charged under the working conditions, and due to the small pore diameters, ion exclusion takes place. In the case of mesoporous titania thin films, the higher isoelectric point implies a lower titanolate surface speciation [51,52]; thus, the negatively charged density was low at a pH around 5 and almost neutral at pH 5.5. No electrostatic exclusion and only little adsorption of the ruthenium complex were registered near the titania isoelectric point. As seen in Figure 5a,b, both probes were able to penetrate through the mesoporous titania and reach the electrode.

Incorporation of PMETAC to silica or titania mesoporous thin films lead to marked changes in the permeation behavior with respect to the pure oxides. In the case of [Ru(NH_3_)_6_]^3+/2+^ (Figure 5c), a practically total exclusion of the probe is observed. This can be explained by the highly positive charge density imparted by the tethered polymer that prevents the positively charged probe from diffusing into the electrode. The observed faradaic currents are significantly lower than those found in PMETAC-modified mesoporous silica thin films obtained through a free radical polymerization route [41]. This fact can be attributed to the external polyelectrolyte layer, which can act as an extra fixed electrostatic barrier that hinders permeation.

The CV measurements of the negatively charged probe [(Fe(CN)_6_]^3−/4−^ show a less evident behavior. In the case of the PMETAC-modified silica, ferri/ferrocyanide is practically excluded and prevented from diffusing into the electrode. This counterintuitive behavior is significantly different from the work of Andrieu-Brunsen et al. that was previously discussed [41], in which [(Fe(CN)_6_]^3−/4−^ is able to diffuse through the PMETAC-functionalized membranes with moderate-to-high polyelectrolyte content. The observed exclusion behavior in our results can be attributed to the different surface functionalization and polymerization techniques employed in this case, which leads in our case to a thicker overlayer and a dense confined polyelectrolyte, which hinders diffusion of [Ru(NH_3_)_6_]^3+/2+^ completely. This scenario presents a high ionic density made up of tethered PMETAC and counterions within the mesopores, which can lead to a sort of zwitterionic pore interior containing highly negative and positive charge density, leading to bipolar exclusion [11,53]. The effect of alternate positive and negative charge layers inhibiting the diffusion of mobile charges has been successfully proven by Piccinini et al. for LBL layers [54]. In the case of the PMETAC-modified titania, negative probes partially permeate at pH 5.5, although the signal intensity decreases to a great extent with respect to the pure oxide sample. This suggests that the bipolar exclusion is linked to the coexistence of the PMETAC moieties and the fixed surface charges, which act as fixed counterions, leading thus to a zwitterionic-like fixed charge environment. In the case of silica, the negative charges are higher due to the greater difference with the isoelectric point. The bipolar exclusion is less important in the case of the titania matrix, which is closer to the isoelectric point. Again, this illustrates that permselectivity can be controlled by controlling the inorganic framework and using its speciation behavior [38,55]. Apart from the bipolar exclusion due to the presence of positive and negative charges coexisting in a confined environment, an additional steric impediment might arise from the polymer presence both in the mesopores and the surface layer in the hybrid systems. The presence of polymer filling drastically decreases the available pore volume. As mesoporous channels became mainly microporous, the steric hindrance could lead to a decrease in all signals. Furthermore, as charged probes were forced to diffuse through a smaller space with higher viscosity, confinement might aid to hinder ion transport.

## 4. Conclusions

This work demonstrates that post-grafting followed by ATRP constitutes a plausible route to grow PMETAC brushes on both mesoporous silica and titania thin films, with similar yields, as can be seen by DRIFT and ellipsometry. This procedure to functionalize both silica and titania paves the way for the right choice of scaffolds when a specific purpose is perused (i.e., titania has proven to be a much better substrate for cell adhesion due to the higher stability in aqueous environments [37], while silica has been widely used for drug delivery purposes [3]). Environmental ellipsometric porosimetry results show a drastic change upon polymer growth in comparison to previous work. Complete pore filling is reflected in the replacement of mesoporosity with a microporous hybrid phase, as revealed by the absence of hysteresis in the EEP measurements. In addition, polymer growth on top of the film surface was unequivocally detected through a double-layer EEP model. We obtained an estimated DP of ~2–3 inside the pore walls, while the polymer layer on top of the mesoporous material reaches a DP between 30–70 and swells up to 2.5–3 times its original thickness upon water intake at higher water vapor pressures. Water adsorption isotherms also demonstrates that pore volume drops from 45% and 38% to 26% and 22% after post-grafting functionalization (silica and titania, respectively) and to approximately 7% after polymerization.

The ATRP polymerization method for pore functionalization presented here leads to a homogeneous distribution of polymer chains and as small polydispersity based on an halogen exchange method which slows down propagation relative to initiation. Dormena tspeciess limit the concentration of the radicals, therefore reduce their coupling, allow for their controlled growth, ensure a uniform polymerization and avoid pore blocking. These features are highly required, as fast polymer growth on the surface would favor pore blocking and high radical concentration in confinement would increase termination rates, therefore leading to shorter and heterogeneous polymer growth in confined environments.

The complete pore filling plus the polymer overlayer presented here open the gate for tuning ion transport phenomena through the control of the polymer–inorganic interactions and location. In principle, a combination of steric hindrance and charge exclusion effects can play an important role in regulating the transport properties of hybrid polymer–inorganic mesoporous thin films. Additionally, this work reinforces the key role of the surface chemistry and speciation of the titania and silica matrices in the overall permselective behavior [11,56]. Interestingly, the cooperative interplay between the polymer brush charge and the matrix isoelectric point gives rise to distinctive permselective behavior, different from the original matrix. The membranes may hinder or allow the transport of positive or negative probes, giving rise to four different possibilities, each of which can be attained by one of the membranes described in this work. These results show the relevance of the inorganic matrix on the permselective properties, which is normally underestimated, and the importance of its choice in the design of permselective membranes, even after functionalization.

Additionally, it is worth calling attention to the combination of high-density grafted cationic polyelectrolytes and negatively charged inorganic mesoporous films and its resemblance to zwitterionic interfaces. Tethering positive charges inside the pore may not only create exclusion but may also end in a double exclusion if both charges are present, as proposed in previous works [11,53]. We hope our results will shed some light on the importance of the matrix in the design of intelligent tailorable materials as the leading player in the properties sought. Furthermore, we provide new experimental data that could be compared with computational simulation advances to provide a better insight into the subject, not only regarding diffusion processes of small molecules through nanometer-sized pores, but also the study of brushes in confinement and their conformations.

## Figures and Tables

**Figure 1 polymers-14-04823-f001:**
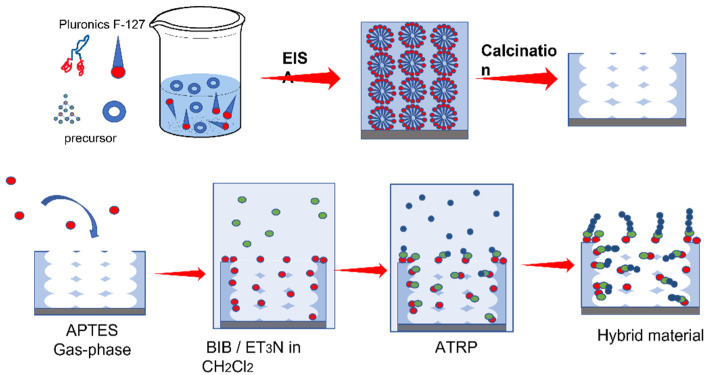
Scheme of the synthesis of the hybrid material. For MTF-TiO_2_, TiCl_4_ was used as a precursor, and TEOS was used for MTF-SiO_2_.

**Figure 2 polymers-14-04823-f002:**
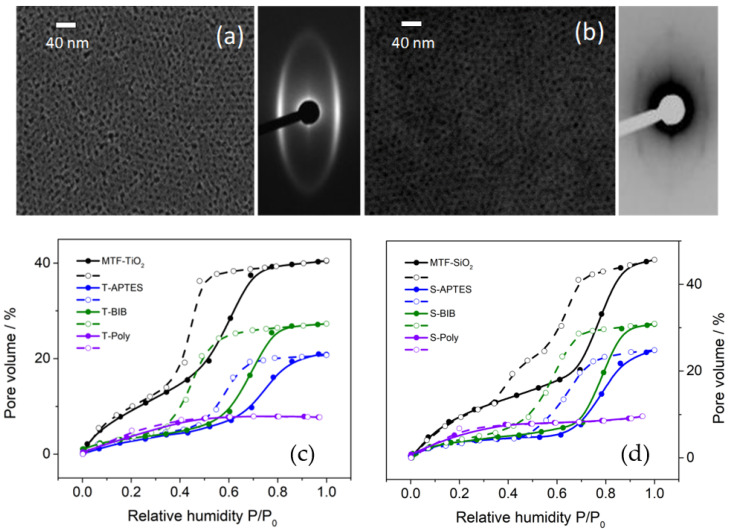
Typical SEM micrographs and SAXs patterns of calcined samples of (**a**) MTF-TiO2 and (**b**) MTF-SiO_2_. Pore volume measured by EEP for MTF-TiO_2_ (**c**) and SiO_2_ (**d**) before and after being functionalized with APTES (T-APTES and S-APTES), APTES-BIB (T-BIB and S-BIB) and APTES-BIB-ATRP (T-Poly and S-Poly).

**Figure 3 polymers-14-04823-f003:**
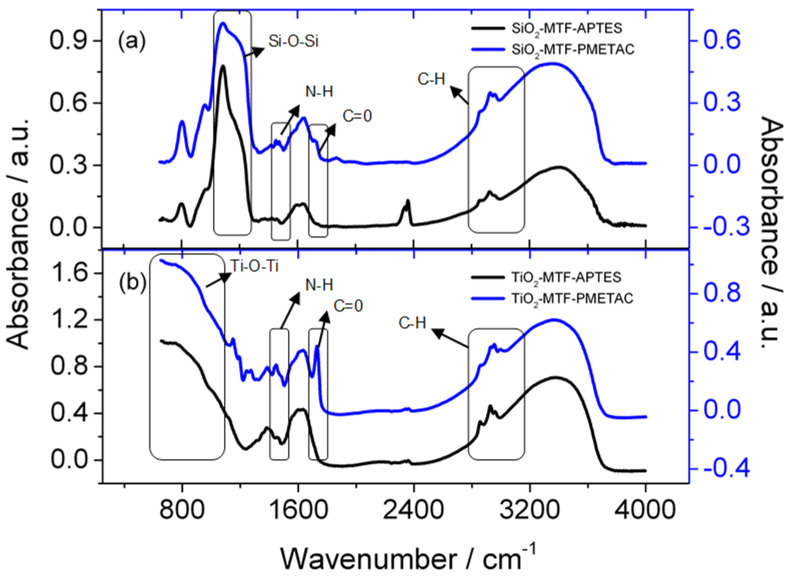
DRIFTS measurements for silica (a) and titania (b) mesoporous thin films before (black) and after polymerization (blue). The wavenumber region of the most relevant absorption bands of selected functional groups have been outline for clarity.

**Figure 4 polymers-14-04823-f004:**
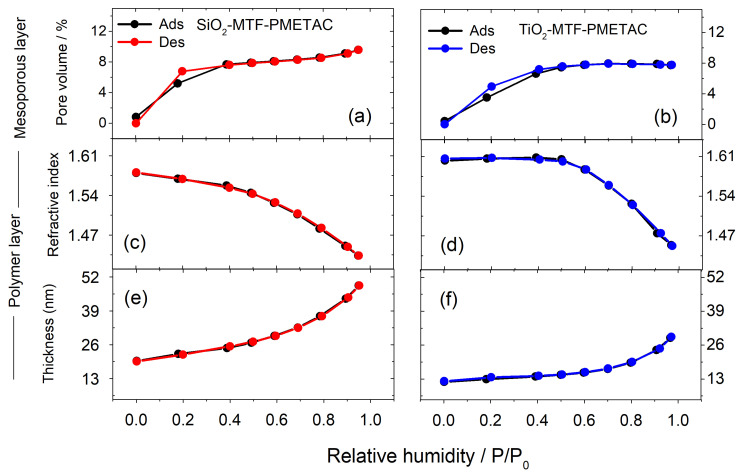
A two-layer model for the EEP measurements: mesoporous layer and a polymer overlayer. Accessible pore volume of the mesoporous silica (**a**) or titania layer (**b**) is shown. The variation of the PMETAC overlayer refractive index for silica or titania systems is given in (**c**) and (**d**), respectively. The PMETAC overlayer thickness for silica or titania systems is shown in (**e**) and (**f**), respectively, for increments of relative humidity.

**Figure 5 polymers-14-04823-f005:**
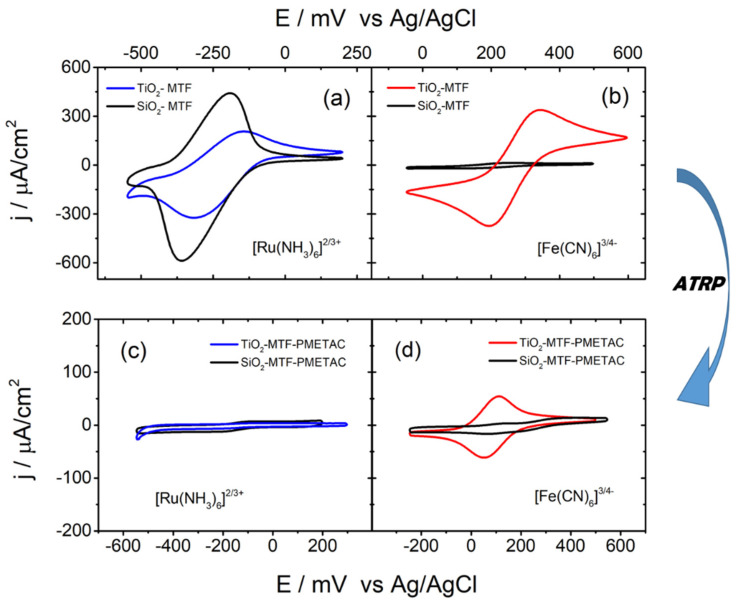
Cyclic voltammetry of [Ru(NH_3_)_6_]^3+/2+^ (**a**,**c**) and [(Fe(CN)_6_]^3−/4−^ (**b**,**d**) for an ITO electrode covered with SiO_2_-MTF and TiO_2_-MTF. Measurements were conducted at pH = 5.5. CV measurements on MTF are shown in (**a**,**b**), and on PMETAC-modified MTF are shown in (**c**,**d**).

## Data Availability

The data presented in this study are available on request from the corresponding author.

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
