# Peer review of "A Comparative Study of PMETAC-Modified Mesoporous Silica and Titania Thin Films for Molecular Transport Manipulation"

_polymers, 2022, doi:10.3390/polym14224823_

Round 1
Reviewer 1 Report
The paper "A Comparative Study of PMETAC-modified Mesoporous Silica and Titania Thin Films for Molecular Transport Manipulation" presents interesting results and can be accepted for publication after major revision. The following issues should be clarified.
Please add an appropriate scheme to illustrate the full synthesis of the hybrid materials.
Please add information about the thicknesses of the APTES and 2-bromoisobutyryl bromide coatings, and what grafting density of the initiating layer. it is a very important parameter for following initiating polymerization. Please add information about the polymerization time.
Can the Authors add information about the molecular mass and grafting density of the grafted brushes, this parameter is more understandable for Readers than the number of monomers per surface area.
There also is a critical remark concerning the organization of ellipsometric studies. The authors mentioned the "correlation problem" between unknown parameters of the thin-film system that leads to increasing errors. It is known that this problem is significantly aggravated for ultra-thin coatings (when the thickness of the film is much smaller than the wavelength), and especially when the surface model is complicated considerably. Obviously, the authors had to face this difficulty in solving the inverse problem of ellipsometry. The appropriate discussion should be added. It is also clear that the authors were not the first among those who sought various approaches to solving this problem. In view of this, I think it is appropriate to mention a number of works where original methods were proposed to reduce the correlation dependence of parameters during ellipsometric measurements:
https://doi.org/10.1364/AO.54.006208
https://doi.org/10.1039/C7SM02285A
Author Response
Reviewer 1
1) Please add an appropriate scheme to illustrate the full synthesis of the hybrid materials.
A new scheme was added for the synthesis of the material as Figure 1.
2) Please add information about the thicknesses of the APTES and 2-bromoisobutyryl bromide coatings, and what grafting density of the initiating layer. it is a very important parameter for following initiating polymerization. Please add information about the polymerization time.
The thickness of the functionalization was measured. Due to the small changes in thickness for these monolayers, to assess this request, a silicon substrate was used. This provides better contrast and similar chemistry due to the silicon oxide passivation layer. The substrate was measured before and after functionalization. A silicon dioxide layer was modeled. A native oxide layer of 2.5+-0.05 nm was measured on the substrate before functionalization, the same layer was measured to be 3.1 +- 0.07 nm after functionalization. If we accept similar refractive indices for the APTES monolayer and the native oxide, a total thickness of 0.6 +- 0.1nm was calculated. The same substrate was functionalized with 2-bromoisobutyryl bromide and a thickness of 0.4+-0.1 was calculated for this layer following the same approach.
We added this information in the text in line 194 “This could also be confirmed by measuring the monolayer of APTES and 2-bromoisobutyryl bromide in a planar surface of silicon, being (0.6 +-0.1) for APTES and (0.4+-0.1nm) for BIB. A comparison of monolayer thickness and the decrease in pore volume, confirms 2.3-3 molecules per nm2 for silica and 1.5-2 for titania. Consequently, a high number of grafting sites for the ATRP initiator (at least 90% of these grafting sites reacted with 2-bromoisobutyryl bromide) and subsequent polymerization were available when compared to co-condensation thin film synthesis”
The polymerization time was added in materials and methods line 121: “Mesoporous substrates were removed from the polymerization solution after 20 hours”
3) Can the Authors add information about the molecular mass and grafting density of the grafted brushes, this parameter is more understandable for Readers than the number of monomers per surface area.
The approach of this work and the limited mass of the synthesized polymers and oligomers restrain the use of GPC for polymer molecular mass. Therefore, the quantitative value “number of monomers per surface area” is reported. If an initiation efficiency of 35 % is accepted as reported elsewhere, a grafting density of 0.8 polymer chain per nm2 can be calculated (also in agreement with previous reports doi.org/10.1021/ma801643r) and the average expected degree of polymerization would range between 2-4 monomer per polymer. These values agree with previous reports and have been studied by Brunsen and the citation has been added to the manuscript (DOI: 10.1021/cm503748d). furthermore, a homogeneous distribution of polymer chains is expected based on a homogenous initiator grafting and as small polydispersity. A halogen exchange method was used which slows down propagation relative to initiation. dormant species and thus limit the concentration of the radicals to reduce their coupling and allow for their controlled growth.
Despite the small mass we are dealing with, our approach provides relevant information on the location of the polymer chain, this has been omitted in most of the publications and discussed in a few. We highlight the limitation in polymerization degree inside pores and the decrease in pore volume after post-grafting (in agreement with the literature). From the Thickness of the top layer, a DP of 30-70 monomers can be estimated. Consistent with reports in porous particles doi.org/10.1021/ma801643r.
The grafting density could be calculated from the pore volume drop after post-grafting compared to the thickness of the monolayer and previously reported monolayer grafting density. (https://doi.org/10.1021/la304719y)
This information was added in line 259 “A decrease in volume fraction of 15% and 13% for silica and titania, respectively, after polymerization allows us to estimate the number of monomers per surface area, being 2.0 nm2 for titania and 2.8 nm2 for silica from molecular volume values taken from the literature and geometrical considerations [24,40,41]. If an initiation efficiency is assumed to be 35 % as reported elsewhere a DP of 2.7 would be expected for both titania and silica [22]. ”
We would like to stress that the aim of this work was not to study grafting density and degree of polymerization as this has been studied previously but to bring a comparison between mesoporous titania and silica as scaffolds. Titania has scarcely been used as compared to silica and here we report how the functionalization works analogously for both scaffolds. The possibility of fictionalized titania in a similar fashion as silica allows us to implement this material in a wide variety of applications where silica may not be optimal, due to solubility or mechanical properties. We further show and compare the transport properties of both hybrid materials showing the way different materials can widen the properties of hybrid thin films. Moreover, we show that these materials may act not only as scaffolds but as an active players in the final properties of the hybrid material.
4) There also is a critical remark concerning the organization of ellipsometric studies. The authors mentioned the "correlation problem" between unknown parameters of the thin-film system that leads to increasing errors. It is known that this problem is significantly aggravated for ultra-thin coatings (when the thickness of the film is much smaller than the wavelength), and especially when the surface model is complicated considerably. Obviously, the authors had to face this difficulty in solving the inverse problem of ellipsometry. The appropriate discussion should be added. It is also clear that the authors were not the first among those who sought various approaches to solving this problem. In view of this, I think it is appropriate to mention a number of works where original methods were proposed to reduce the correlation dependence of parameters during ellipsometric measurements:
To try to avoid correlations, and minimized the number of parameters, sequential and independent measurements were done on ITO on silica. These parameters were kept constant for the subsequent measurements. To assess the significance of the measurements several things have been considered, in detail in the work a two and a single-layer model was tested after polymerization. For the two-layer model, the thickness of the mesoporous layer was kept constant to minimize parameters. Furthermore, a layer filled with polymer but without any polymer brush was compared.
Additional information was added in line 297: “Correlation between parameters for layers below 20 nm has already been a challenging subject even on standard substrates. In this work, an additional porous layer increases the complexity of the model. To minimize the number of parameters, the substrate (ITO on glass) was previously modeled and its parameters were kept constant. The mesoporous layer thickness was fixed due to minimal expected changes with humidity in microporous hybrid material. To give further support to the results additional measurements were performed for comparison. Figure S1 shows the EEP curves obtained using a single-layer model for the hybrid film.”
The measured data were compared to the literature to check their significance. Such as the thickness and refractive index of the layers as their response to humidity. It is also important to notice that similar results were obtained with two different materials and in a wide range of humidity for each case. Despite not being capable of guaranteeing a complete correlation-free measurement as stated in the text, the overall behavior and values seem in agreement with other simple reports on surface ATRP polymerization and the materials used. And supported by control measurements described in the manuscript and added in the supporting information.
The suggested references were added in the manuscript as references 45 and 46: https://doi.org/10.1364/AO.54.006208 https://doi.org/10.1039/C7SM02285A
Reviewer 2 Report
This manuscript reports the modification of mesoporous silica and mesoporous titania thin films using surface-Initiated Atom Transfer Radical Polymerization (SI-ATRP) of (2-methacryloyloxy)-ethyltrimethylammonium chloride. The ion permeabilities of the modified mesoporous materials were also studied. This manuscript can be recommended to be published in this journal after a major revision. The main concerns are as follows:
1. Figure 1c and d should be the nitrogen adsorption-desorption curve of SiO2-APTES and TiO2-APTES. The authors should give the nitrogen adsorption and desorption curves before and after the modification of mesoporous SiO2 and TiO2, and compare them. The abscissa should be changed to Relative pressor.
2. The authors should give TEM, SEM images of mesoporous SiO2 and TiO2 before and after modification, and compare the morphology.
3. The formation mechanism of mesoporous SiO2, TiO2, the APTES modified mechanism of mesoporous material, and the formation mechanism of PMETAC molecular brush by ATRP should be given. For modification and polymerization mechanism of mesoporous SiO2, TiO2, please refer to: RSC Adv. 2016, 6: 6551,Solid State Sci. 2016, 52: 154,Colloid Surface A, 2015, 481,Soft Matter, 2014, 10(8), 1110,Macromol Chem Phys, 2013, 214(14): 1602
4. The 1H NMR spectrum PMETAC and TGA of MTF-SiO2 and MTF-TiO2 should be given.
Author Response
1) Figure 1c and d should be the nitrogen adsorption-desorption curve of SiO2-APTES and TiO2-APTES. The authors should give the nitrogen adsorption and desorption curves before and after the modification of mesoporous SiO2 and TiO2, and compare them. The abscissa should be changed to Relative pressor.
Old Fig. 1 (now Fig. 2) was replaced. All isotherms after each functionalization step are added.
Finally, the isotherm after polymerization is shown later in the text and discussed accordingly. The Abscissa is relative humidity since the pressure is kept constant and only the amount of humidity is changed. The technique used is environmental porosimetry at room temperature and not N2 adsorption.
2) The authors should give TEM, SEM images of mesoporous SiO2 and TiO2 before and after modification, and compare the morphology.
After functionalization, the organic film of a few nm was not possible to be seen by cross-section at SEM due to the charging effect and/or sample burning with scanning.
3) The formation mechanism of mesoporous SiO2, TiO2, the APTES-modified mechanism of mesoporous material, and the formation mechanism of PMETAC molecular brush by ATRP should be given. For modification and polymerization mechanism of mesoporous SiO2, TiO2, please refer to: RSC Adv. 2016, 6: 6551,Solid State Sci. 2016, 52: 154,Colloid Surface A, 2015, 481,Soft Matter, 2014, 10(8), 1110,Macromol Chem Phys, 2013, 214(14): 1602
A scheme was added for the synthesis of the material as Figure 1.
Reference “Macromol Chem Phys, 2013, 214(14): 1602” was added as reference 43
4) The 1H NMR spectrum PMETAC and TGA of MTF-SiO2 and MTF-TiO2 should be given.
The sample discussed here is a thin film and no NMR can be performed, since the polymerization occurs at the surface, and very little mass is under consideration (200 nm thick film in a 1cm square footprint). We agreed with the reviewer that TGA could give an idea of the total polymer amount and would be meaningful to support our description but would not provide more details as it won’t be able to distinguish the location of the polymer, as we have described in our work.
Round 2
Reviewer 1 Report
The authors put a lot of effort into this manuscript, which is now perfect and ideally suited for publication.
Reviewer 2 Report
The revised manuscript can be accepted.